https://doi.org/10.1038/s42003-020-01321-5　**OPEN**

# Cryo-electron tomography of cardiac myofibrils reveals a 3D lattice spring within the Z-discs

Toshiyuki Oda [1✉] & Haruaki Yanagisawa[2]

The Z-disc forms a boundary between sarcomeres, which constitute structural and functional units of striated muscle tissue. Actin filaments from adjacent sarcomeres are cross-bridged by α-actinin in the Z-disc, allowing transmission of tension across the myofibril. Despite decades of studies, the 3D structure of Z-disc has remained elusive due to the limited resolution of conventional electron microscopy. Here, we observed porcine cardiac myofibrils using cryo-electron tomography and reconstructed the 3D structures of the actin-actinin cross-bridging complexes within the Z-discs in relaxed and activated states. We found that the α-actinin dimers showed contraction-dependent swinging and sliding motions in response to a global twist in the F-actin lattice. Our observation suggests that the actin-actinin complex constitutes a molecular lattice spring, which maintains the integrity of the Z-disc during the muscle contraction cycle.

---

[1] Department of Anatomy and Structural Biology, Graduate School of Medicine, University of Yamanashi, 1110 Shimokato, Chuo, Yamanashi 409-3898, Japan. [2] Department of Cell Biology and Anatomy, Graduate School of Medicine, the University of Tokyo, 7-3-1 Hongo, Bunkyo-ku, Tokyo 113-0033, Japan. ✉email: toda@yamanashi.ac.jp

The Z-disc defines the boundary between the two adjacent sarcomeres by crosslinking the two anti-parallel thin myofilaments. The Z-discs transmit tension generated by muscle contraction through the crosslinks mainly composed of α-actinin[1,2]. The α-actinin belongs to the spectrin family and forms a rod-shaped antiparallel homodimer of length ~35 nm. The α-actinin monomer has the N-terminal actin-binding domain, which is composed of two calponin homology (CH) domains; the central rod domain, which is composed of four spectrin-like repeats; and the C-terminal tandem EF-hand domains, which are insensitive to calcium in the muscle-type isoforms[3–5].

The structure of vertebrate Z-disc has been intensively studied using conventional ultra-thin section electron microscopy for decades[6–11]. In the transverse sections, the Z-discs show two distinct structural states: the "small square" form and the "basket-weave" form. In the classical studies of fixed muscle tissues, it has been proposed that the small square and the basket-weave forms represent the relaxed and the active contracted states, respectively[7,11,12]. According to the recent observations, however, isolated myofibrils immersed in relaxing solution containing EGTA and ATP exhibit the basket-weave form[13,14]. Considering the previous observation that non-treated muscle tissues show mixture of the small square and the basket-weave forms within a single Z-disc[8], the variable ionic environment within the cytoplasm of the muscle tissue specimen makes it difficult to obtain a regular lattice structure of the Z-discs[15]. In this study, therefore, we performed cryo-electron tomography of isolated myofibrils in the presence of either EGTA + ATP or calcium+ATP (Ca+ATP). Under the ion-controlled conditions, we visualized the 3D structural changes within the Z-discs.

## Results

**Thin-section microscopy of cardiac myofibrils.** Before conducting the cryo-electron microscopy, we observed the isolated myofibrils using conventional ultra-thin section electron microscopy (Supplementary Fig. 1a) because the structure of the Z-disc in the Ca + ATP-activated state has not been reported. Although the Z-discs in the relaxed (EGTA + ATP) state showed a square lattice with inter-actin connections (Fig. 1a, c), the Z-discs in the Ca + ATP state exhibited a diamond-shaped lattice with inter-axial angles of 80° and 100° (Fig. 1b, d). When we carefully examined the previous studies, however, the reported "small square" lattice is not always square and diamond-shaped "offset-square" lattices have often been reported in mammalian Z-discs[8,9,16]. Thus, it is possible that the Z-discs under the Ca + ATP condition are in a fully activated state, which takes a diamond-shaped lattice rather than a square lattice. We acquired tomograms of ultra-thin sections of the myofibrils and used the averaged subtomograms for the initial references in the subsequent cryo-electron tomography.

**Cryo-electron tomography of native cardiac myofibrils.** Cryo-electron tomography of myofibrils has been challenging due their large size (diameter of ~2 μm). However, we happened to notice that cardiac myofibrils are often branched into thin (300–500 nm) sub-fibrils when intervened by mitochondria or nuclei[17]. We purified these "thin" myofibrils by gentle homogenization and differential centrifugation and conducted cryo-electron tomography to reconstruct the repeating unit within the Z-discs (Fig. 2a–d). We compared the sarcomere lengths in the EGTA + ATP and the Ca + ATP states and confirmed that the Ca + ATP treatment properly activated the myofibrils[18] (Supplementary Fig. 1b). Lattice points of the F-actin were clearly observed in the cross-sections of the Z-discs (Fig. 2e). We extracted subtomograms using these lattice points as centers. The resulting averaged subtomograms (Fig. 3a, b and Movie S1) showed a

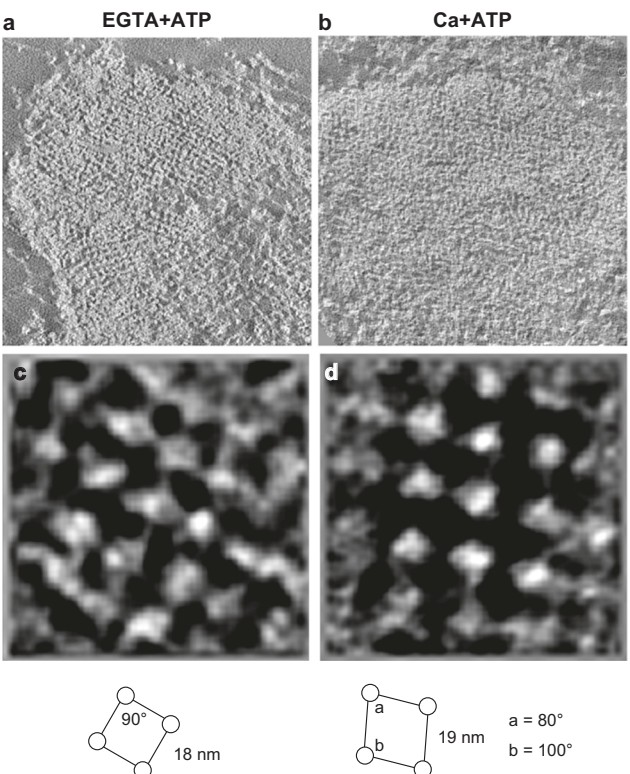

**Fig. 1 Electron microscopy of epon-embedded cardiac myofibrils. a**, **b** Slices of tomograms showing the cross-sections of the Z-discs in the EGTA + ATP and the Ca + ATP states. Black and white of the images were inverted. **c**, **d** Slices of averaged subtomograms showing the actin lattices. The square lattice of the EGTA + ATP state and the diamond-shaped lattice of the Ca + ATP state were shown.

central F-actin (gray), four opposite-polarity F-actins (yellow), and eight α-actinins (orange). Due to the structural heterogeneity within the actin lattice, however, other neighboring F-actins were blurred out. Therefore, we shifted the center of the subtomograms from the central F-actin to the neighboring F-actins and conducted additional alignments to generate shifted maps. By combining six shifted maps with the original map (Fig. 3c, d), we could see the actin lattice in the basket-weave form (EGTA + ATP) and the diamond-shaped form (Ca + ATP) (Fig. 3e, f). When we aligned the shifted maps with the original map, we noticed that densities of the α-actinin dimers were not fully visualized in the original map (Fig. 3a, b, orange), and the shifted maps showed the remaining lower halves of the dimers (Fig. 3e, f, green). These observations suggest that there is a structural heterogeneity within the α-actinin dimer and the 3D refinement was biased toward one of the α-actinin molecules that binds to the F-actin at the center of the subtomogram.

We attempted to improve the resolutions of the actin-actinin complexes by focused refinement, but we could not align the F-actin and the α-actinin densities simultaneously, suggesting that the interaction between the F-actin and the actin-binding domain of the α-actinin is highly flexible[19,20]. Thus, we performed multibody refinements[21], which aligned the central F-actin (gray), the opposite-polarity F-actin (yellow), and two monomers of the α-actinin dimer (orange and green), separately (i.e. four-body refinement). We repeated this multibody refinement for all the four anti-parallel F-actin pairs, and reconstructed the whole repeat unit of the Z-disc with ~23 Å (Supplementary Fig. 1c and Fig. 4a, and Supplementary Movie S2), which was improved compared to the previous results (39 Å ref. [14] and 58 Å ref. [22]). The resolutions of the α-actinin

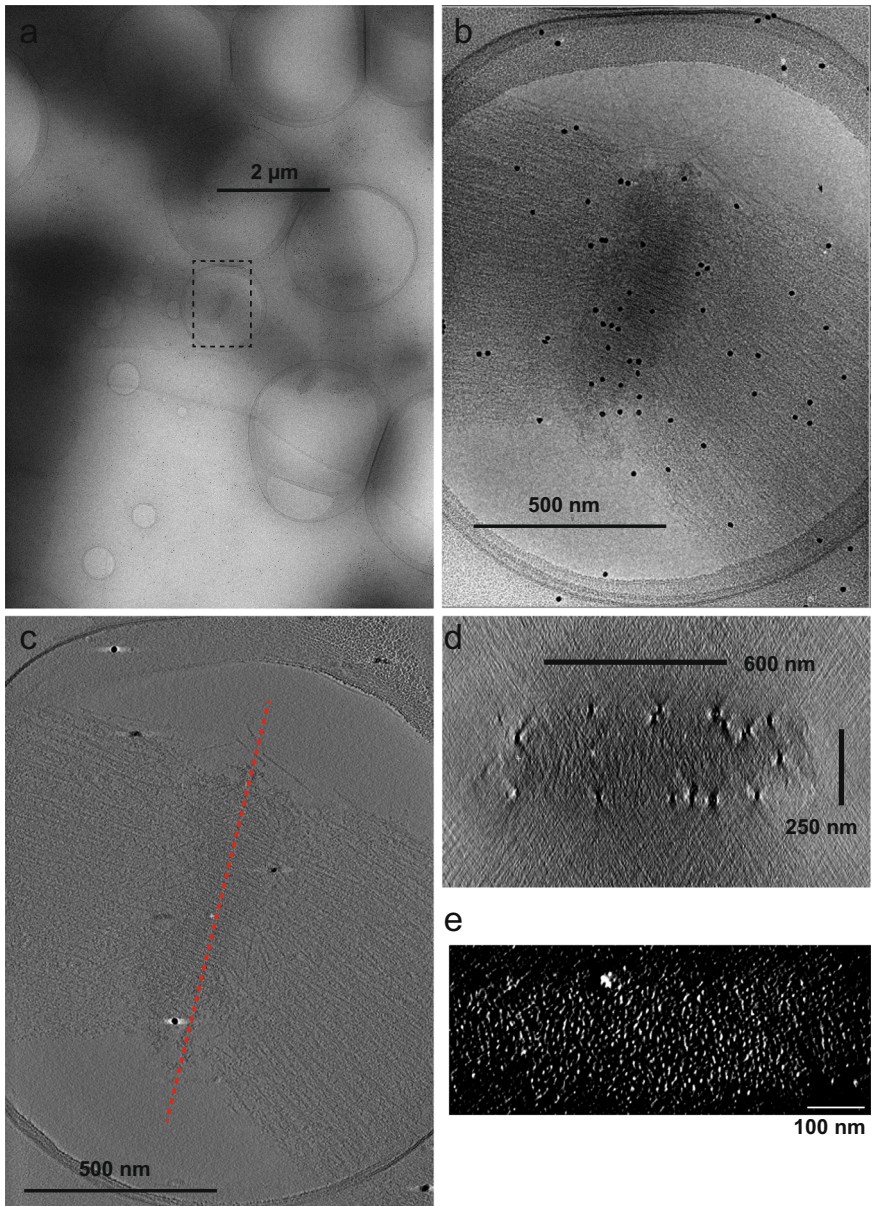

**Fig. 2 Cryo-electron microscopy of native cardiac myofibrils. a** Low-magnification image of a thin myofibril in the EGTA + ATP state. Broken box indicates the recording area in **b**. **b** 0° tilt image of the Z-disc. **c** Slice view of the reconstructed tomogram. Red broken line indicates the position of the cross-section in **d**. **d** Cross-section of the tomogram of the Z-disc. **e** Lattice points visualized by enhancing the contrast of the cross-sectional view. These lattice points were manually picked for extraction of the subtomograms.

densities were sufficient for rigid-body docking of the crystal structure[3] (Fig. 4a, red and green models).

In both EGTA + ATP and Ca + ATP states, nearly 40% of the variance was explained by the first and the second eigenvectors (Figs. 4b, S1d, and Supplementary Movie 3). In both states, the first eigenvector represents a tilting motion of the central F-actin about its center. This motion can be related to the presence of the flexible linker between the N-terminal actin-binding domain and the rod domain of the α-actinin[3]. The second eigenvector in both states represents a rotation of the opposite-polarity F-actin about the central F-actin. This eigenvector probably reflects the irregularity in the lattice angles[9].

**Contraction-induced lattice conversion in the Z-disc.** The assembled refined bodies showed large structural changes between

the EGTA + ATP and the Ca + ATP states (Fig. 5a–c, Supplementary Movies 4 and 5). The opposite-polarity F-actin was displaced by ~80 Å along with a ~13° turn about its long axis. It has been reported that myosin motors apply a right-handed torque to the F-actin[23]. The axial rotation observed in our maps is likely to be related to this myosin-dependent screw rotation. In response to the F-actin rotation, the α-actinin dimers showed a ~21° swinging motion about the central F-actin along with an intra-dimer sliding. The conformation of the α-actinin in the EGTA + ATP state was similar to the proposed model of the basket-weave form[8,14,19]. Although it is difficult to compare the previously reported 3D structure of the "small square" Z-disc[24] with our Ca+ATP map due to the limited resolution of the previous thin-section tomogram, the straightened conformation of the α-actinin in the top view is consistently observed in both observations. These motions of the α-actinin caused sliding of the actin-binding domain along

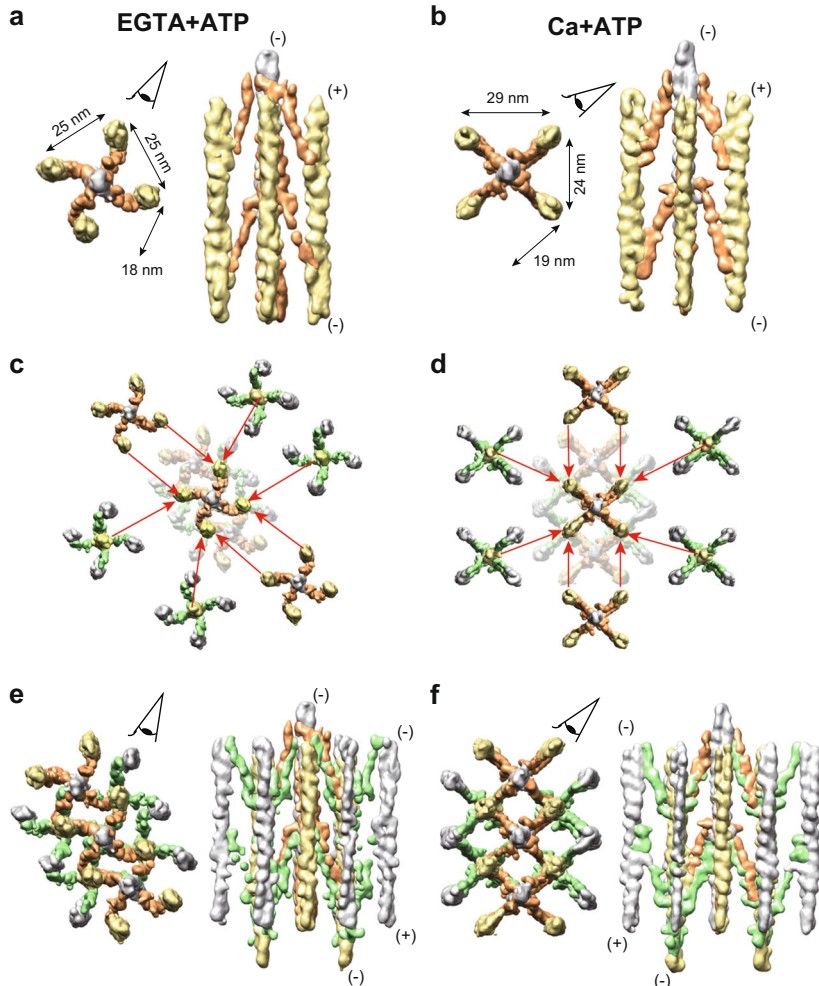

**Fig. 3 Subtomogram averaging of the repeat unit of the Z-disc. a, b** Averaged subtomograms composed of the central F-actin (gray), the opposite-polarity F-actins (yellow), and the α-actinin molecules (orange). Eye symbols on the top views indicate the direction of the side views on the right. (+): barbed end; (−): pointed end. Top views are seen from the pointed end of the central F-actin. **c, d** Combination of the six shifted maps with the original map. Coordinates were re-centered to the neighboring six F-actins (four opposite polarity F-actins, and two parallel F-actins) and additional 3D refinements were conducted based on the new coordinates. The resulting shifted maps were then superimposed by aligning one F-actin of the shifted map with the corresponding F-actin of the original map, as indicated by the red arrows. **e, f** Composite maps created by combining seven maps. The EGTA + ATP and the Ca + ATP states show the basket-weave and the diamond-shaped lattice conformations, respectively. Both monomers of the α-actinin dimer (orange and green) were fully visualized in the combined maps. Eye symbols on the top views indicate the direction of the side views on the right.

the surface of the F-actin (Fig. 5c, right). Together with the first eigenvector motions (Fig. 4b), it is likely that the interface between the F-actin and the actin-binding domain of the α-actinin is not strictly defined.

## Discussion

In this study, we reported the first native 3D structure of the mammalian Z-disc. The 3D structure of the native invertebrate Z-disc has previously been reported[22], but the invertebrate Z-discs are separated from the actomyosin system under a harsh condition (extraction using 0.6 M potassium iodide). Moreover, the reconstructed map severely suffers from the missing wedge of information because the isolated invertebrate Z-discs are uniformly oriented perpendicular to the beam axis. In our recording condition, the F-actin lattices were randomly oriented relative to the beam axis, and thus the missing wedges were fully compensated by subtomogram averaging. Therefore, our observations were physiologically more relevant for analyzing the conformational changes of the Z-disc in the context of active myofibrils.

Although it has been accepted that the Z-disc takes a square lattice conformation irrespective of whether it is in the small square or in the basket-weave form, diamond-shaped lattices were observed in unstimulated rat skeletal muscle (inter-axial angles of 80°/100°)[25], in dog cardiac muscles (inter-axial angles of 82°/98°)[8], and in the nemaline rods of human myopathy patients (inter-axial angles of 75°/105°)[9]. As the thin and thick myofilaments in the A-band constitute a hexagonal lattice[7], it is possible that tension applied by the actomyosin contraction deforms the square lattice of the Z-discs into the diamond-shaped lattice.

It has been demonstrated that tetanized muscle tissue induced by high-frequency electrical stimulation shows the Z-discs in the basket-weave form[6,16]. This observation contradicts our results and other reports that the basket-weave form is observed in the presence of EGTA and ATP[13,14]. The physical state of the Z-disc in the tetanized muscle may be different from that under the normal contracting condition. As reported in the previous thin section tomography study of the Z-disc in the small square form[24], the actinin cross-bridges were formed between F-actins of opposite polarity in both Ca + ATP and EGTA + ATP states

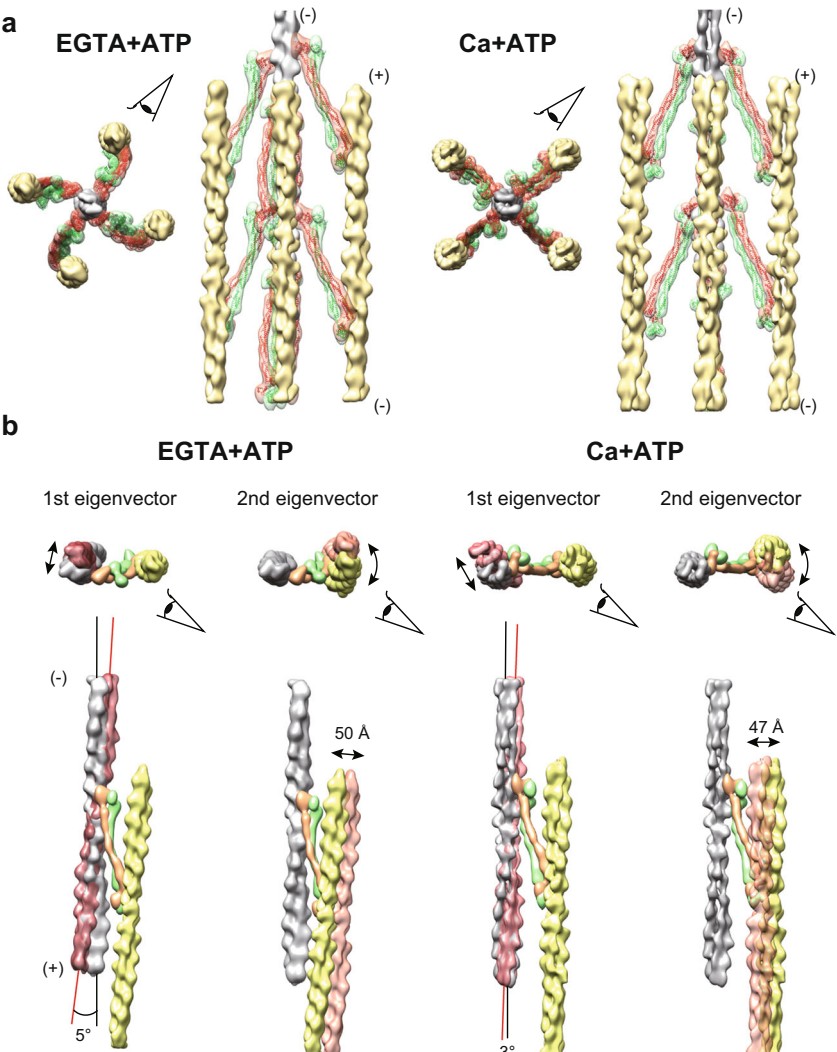

**Fig. 4 Multibody refinement of the F-actin and the α-actinin. a** Composite maps composed of one central F-actin (gray), four opposite-polarity F-actins (yellow), sixteen α-actinin monomers (orange and green). The α-actinin crystal structures (PDB 4D1E) were fitted into the α-actinin maps (red and green models). Eye symbols indicate the direction of the side views on the right of the respective top views. (+): barbed end; (−): pointed end. **b** Visualization of the first and the second eigenvectors (Supplementary Fig. 1d). In both states, the first eigenvector represents tilting of the central F-actin about the center of the filament, and the second eigenvector represents rotation of the opposite-polarity F-actin about the central F-actin.

(Fig. 3). A similar model has been proposed as a "diagonal square" form[26–28]. It is possible that our "diamond-shaped" form corresponds to, or related to the "diagonal square" form, which is thought to be different from the small square form. It would be necessary to conduct a cryo-FIB tomography[29,30] of native muscle fibers to make a clear distinction among these forms.

Given that the basket-weave form is observed in the relaxed state, the diamond-shaped form is thought to be under mechanical stress imposed by the sliding between the thin and thick filaments. Thus, it is reasonable to assume that the actin-actinin cross-bridging complex constitutes a 3D lattice spring, which goes back and forth between the low-energy basket-weave form and the high-energy diamond-shaped form depending on the contractile state. The flexibility of the actin-actinin lattice spring is likely to depend on the flexibility of the α-actinin, which was visualized as swinging and sliding motions (Fig. 5a, b and Supplementary Movies 4 and 5). The swinging motion of the α-actinin can be attributed to the presence of its flexible linker between the N-terminal actin-binding domain and the central rod domain, which mediates the actin-actinin interaction in various orientations[3,19,20].

Note that we did not perform flexible fitting of the crystal structure to our maps due to the limited the resolution. Therefore, the fitted models were not necessarily accurate, which can be discerned that there are clashes between the EF-hand and the neck region and apparent dissociation between the spectrin repeats (Supplementary Movies 4 and 5)[31]. In addition, the sliding motion between the rod domains of the α-actinin is thought to be dan artifact of rigid-body docking of the crystal structure. The rod domains are supposed to be rigid and inflexible. It is necessary to improve the resolution up to ~8 Å to precisely assign the domains of the α-actinin.

It is uncertain if the hydro-frozen myofibrils observed in this study were actually under tension because the myofibrils were extracted from cells and were not anchored to glass needles, which are conventionally used for force measurements[32]. However, we believe that the myofibrils were supposed to be under tension at the moment of pluge-freezing because most parts of these fibers were pressed against the carbon membrane, while the imaged regions were suspended within the water film encompassing the holes in the carbon membrane (Fig. 2). Measurement of the sarcomere length partly supported this conclusion

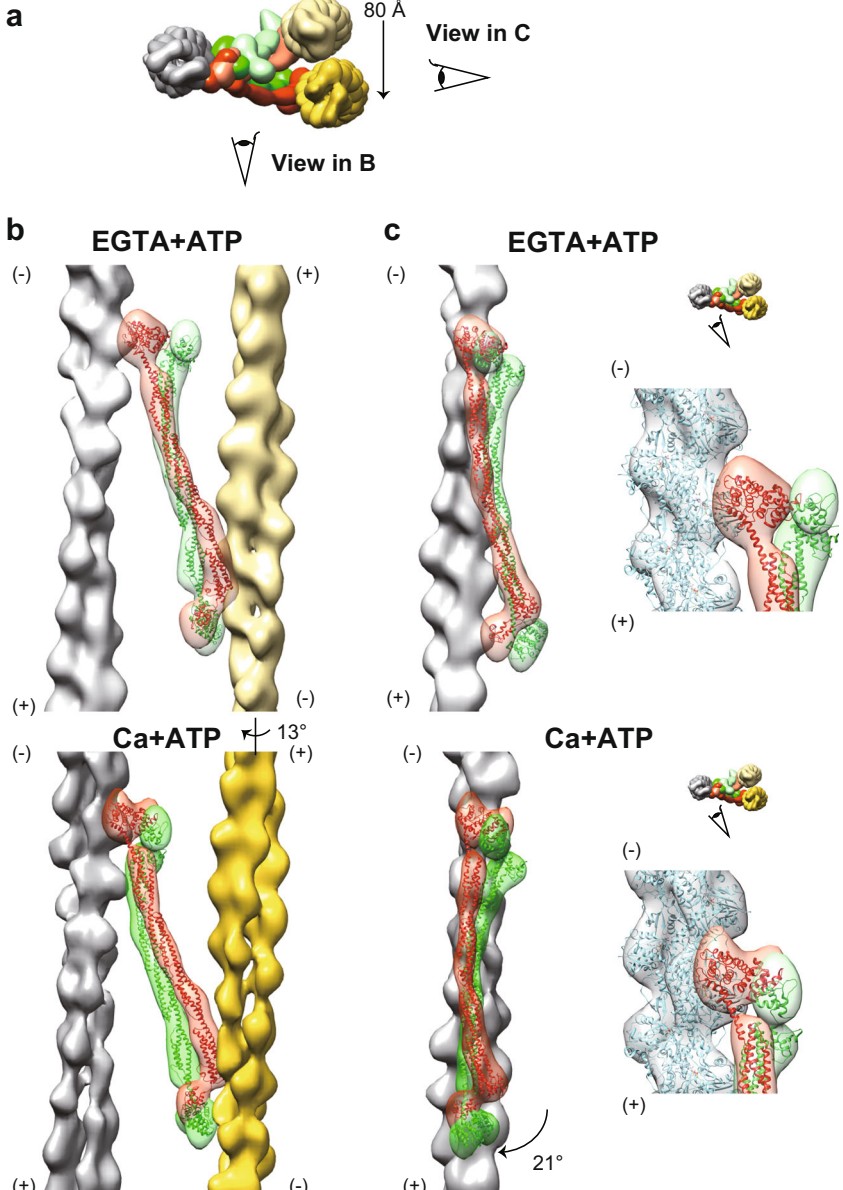

**Fig. 5 Conformational changes in the α-actinin. a** Top view of the actin-actinin cross-bridging complex. Gray: central F-actin; yellow: opposite-polarity F-actin; orange and green: α-actinin monomers. The maps of the EGTA + ATP and the Ca + ATP states are light- and deep-colored, respectively. Eye symbols indicate the direction of the side views in **b**, **c**. **b** The α-actinin dimer twisted about its long axis and the opposite-polarity F-actin showed a ~13° turn. The α-actinin dimer showed a ~21° swing, which translocates the opposite-polarity F-actin by ~80 Å. (**c**, insets) Close-up views of the interface between the central F-actin and the actin-binding domain of the α-actinin. Eye symbols indicate the direction of the close-up views. The F-actin models (PDB 6KLL) were fitted into the central F-actin maps.

(Supplementary Fig. 1b). We further examined the state of the myofibrils by averaging the thin filaments in the A-band (Supplementary Fig. 2). In the EGTA + ATP state, the myosin heads were completely dissociated from the thin filament. Both in the Ca-only and the Ca+ATP states, by contrast, the myosin heads fully decorate the thin filament. In the Ca+ATP state, however, the structure was partially disordered, especially the F-actin region, probably due to the active power strokes of the myosin and resulting heterogenous spatial relationship between the myosin head and the F-actin. These observations support that the myofibrils were properly activated by the addition of Ca and ATP.

The one-start helix of F-actin has a helical parameter of 27.5-Å rise and 166.6°-rotation, giving an ~90°-rotation per seven actin monomers[12,14]. Although this innate 90°-rotation within

the helical symmetry of the F-actin matches the square lattice model (Fig. 6, asterisks), there are large angular discrepancies at other positions. We suppose that these discrepancies between the helical symmetry and the lattice angles underlies the existence of the two lattice forms in the Z-disc. The one-start helix of F-actin can be considered as a two-start helix with 55 Å rise and −26.8°-rotation. If we focus on one strand of this two-start long-pitch helix, two α-actinin dimers bind to the F-actin every seven actin monomers (Fig. 6, red/blue)[14]. As seven is an odd number, the distance between the two longitudinally-neighboring α-actinin dimers was not constant and there are two patterns; 165 Å-rise and −80.4°-rotation, and 220 Å-rise and 107.2°-rotation (Fig. 6, stars). We believe this is another innate feature of the helical symmetry of F-actin, which gives rise to the diamond-shaped

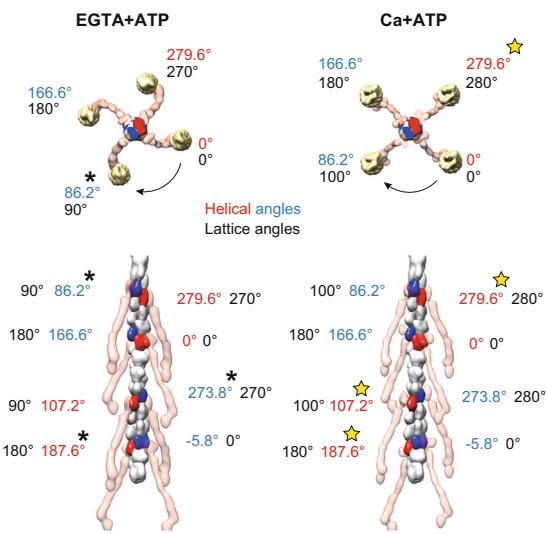

**Fig. 6 Angular mismatches between the helical and the lattice symmetry.** The rotation angles of the F-actin helix and the square/diamond-shaped lattice of the Z-disc were compared. The α-actinin molecules (transparent red) regularly bind to the central F-actin. The α-actinin-bound actin monomers are colored red or blue. The one-start helix of the F-actin with 27.5 Å-rise and 166.6°-twist can be regarded as a two-start helix with 55 Å-rise and –26.8°-twist. The "helical angles" are the rotation angles calculated from the helical symmetry of the F-actin relative to one of the actin monomers designated as "0°". The "lattice angles" correspond to the rotation angles of the opposite polarity F-actins around the central F-actin. In the EGTA + ATP state, the lattice angles are an arithmetic progression with common difference of 90°. In the Ca + ATP state, by contrast, the rotation between the two-neighboring opposite-polarity F-actins is either 80° or 100°. Although the square lattice favors the one-start helix model because of the nearly 90°-turn per seven monomers (asterisks), there are discrepancies in other positions (e.g. 90° versus 107.2°). On the other hand, the diamond-shaped lattice favors the two-start helix model (yellow stars).

lattice with 80°/100° rotation angles in the Ca + ATP state. Note that the basket-weave square lattice with 90°/90° rotation angles is considered to be structurally favorable over the diamond-shaped lattice because the basket-weave form has been observed in relaxed/unstrained state. The variation in the helical twist angles of F-actin[33] could compensate the discrepancy between the square lattice and the helical symmetry of F-actin. In addition, the observed high flexibility between the F-actin and the actin-binding domain of the α-actinin (Fig. 5c, right) could resolve the angular mismatches between the helical and the lattice angles. Although it is counter-intuitive that the stout structure of the Z-disc is maintained by such flexible interactions, the structural plasticity of the α-actinin is thought to be essential for the muscle tissue to withstand the high mechanical stress during the contraction cycles.

The highest resolution achieved in this study was 23 Å, which is insufficient to conduct a flexible docking of crystal structure. Cryo-electron tomography of thick specimen is challenging due to several factors that affect attainable resolution[34]. According to the Crowther criterion $d = \pi \times D/m$ (resolution, $d$; particle diameter, $D$; and number of images, $m$)[35], the resolution limit of a single tomogram is estimated to be 157 Å. In the case of Z-disc tomography, we could overcome this limit by subtomogram averaging because the F-actin lattices were randomly oriented relative to the beam axis. At high angle tilts, however, increase in the inelastic scattering decreases the signal-to-noise ratio. We utilized the energy filter to reduce the noise and the Volta phase plate to enhance the contrast[36], and these instruments did contribute to the improvements in the image quality. Another issue with thick specimen is radiation-induced distortion[37]. In the case of isolated particles such as viruses, the distortion can be corrected by MotionCor2 and by per-particle refinement[38–40]. In the case of filamentous specimen such as myofibrils and axonemes[41,42], however, the structural continuity often complicates the distortion correction and makes per-particle refinement difficult. It would be necessary to develop a new refinement algorithm that is fully compatible with tomograms of filamentous specimen. To improve the resolution of the myofibril tomograms, therefore, it is necessary to reduce the thickness. We found that there is small number of myofibrils that are <200 nm thick. However, such very thin myofibrils were rare and fragile, and they were easily disintegrated by centrifugation and pipetting. Another possible method is cryo-FIB milling of myofibril or high-pressure-frozen muscle fibers[29,43]. Considering that Z-discs of skeletal muscles are more structurally ordered than those of cardiac muscles[14], obtaining 100–200 nm thick lamella from skeletal myofibrils using milling will be a good alternative to improve the resolution of myofibril and Z-disc tomograms.

## Methods

**Isolation of myofibrils.** Fresh porcine hearts were obtained from a local slaughter house. Cardiac myofibrils were dissected from the left ventricular papillary muscles and homogenized in a hypotonic buffer (Hepes 30 mM, pH 7.2) supplemented with protease inhibitor cocktail (Nacalai tesque, Kyoto, Japan) using a kitchen juice mixer. The homogenized tissue was resuspended in a hypotonic HK buffer (Hepes 30 mM, pH 7.2, 60 mM KCl, 2 mM MgCl$_2$, protease inhibitor cocktail) and washed extensively by centrifugation at $750 \times g$ for 15 min at 4 °C until the upper half of the pellet turned white. The brown-colored lower half of the pellet containing collagen fibers were carefully removed and the upper white part was incubated in HK buffer plus 1% Triton X-100 for 15 min at 4 °C. The demembranated myofibrils were washed three times with HK buffer and were gently homogenized with a Dounce glass homogenizer to release thin myofibrils. Large fibrils were removed by centrifugation at $750 \times g$ for 15 min at 4 °C and thin fibrils in the supernatant were collected by centrifugation at $2500 \times g$ for 15 min at 4 °C.

**Ultra-thin section electron microscopy of fixed myofibrils.** The pellet of large myofibrils was re-suspended in HK buffer and treated either with 2 mM CaCl$_2$ and 2 mM ATP or 2 mM EGTA and 5 mM ATP for 5 min at room temperature. The fibrils were then fixed with 1% glutaraldehyde for 1 h at 4 °C and stained with 1% osmium tetroxide and subsequently with 1% uranium acetate. After dehydration in ethanol and acetone, the samples were embedded in Quetol 812 resin (Nissin EM, Tokyo, Japan). Ultrathin sections (60-nm or 200-nm thick) were cut using a ULTRACUT microtome (Reichert Leica) and mounted onto Formvar-coated copper grid. For placing fiducial markers, 15-nm gold particles (BBI Solutions, Cardiff, UK) were attached to both surfaces.

Images were recorded using a JEM-2100F microscope (JEOL, Tokyo, Japan) at University of Yamanashi operated at 200 keV equipped with a F216 CMOS camera (TVIPS GmbH, Gauting, Germany). The nominal magnification was set to ×15,000 with a physical pixel size of 8.6 Å/pixel. Tilt series images were recorded using EM-TOOLs program (TVIPS). The angular range of the tilt series was from −60° to 60° with 2.0° increments and the target defocus was set to 2 μm. Back-projection and subtomogram averaging were conducted as described below in the cryo-EM section. For the subtomogram averaging of the thin-section tomograms, one of the subtomograms was used for the initial reference. Although the epon-embedded specimens were likely to be deformed by shrinkage of the section[14], we believe that it did not affect the averaging of the cryo tomograms because the section-derived maps were used only in the first iteration and subsequently replaced with newly generated averages of cryo tomograms.

**Cryo electron microscopy of native myofibrils.** The pellet of thin myofibrils was resuspended in HK buffer containing cytochrome *c*-stabilized 15-nm colloidal gold[44], and the protein concentration was adjusted to 0.02–0.05 mg/ml. In all, 4 μl of the sample was mounted on freshly glow-discharged home-made holey carbon grids and 1 μl of HK buffer containing either 2 mM CaCl$_2$ and 5 mM ATP or 2 mM EGTA and 5 mM ATP was applied to the sample. Grids were incubated for 1 min at room temperature, blotted for 10 s at 4 °C under 99% humidity, and plunge frozen in liquid ethane using Vitrobot Mark IV (Thermo Fisher Scientific, Waltham, MA). We applied a small offset to the positions of the blotting arms so that one of the filter papers reached the grid slightly earlier than the other, and thus

most of the buffer was blotted from the back-side of the grid. This back-side blotting increased the filament density on the grid. Grid quality was examined using a JEM-2100F microscope. Grids frozen under optimal conditions were then used in the recording session.

Images were recorded using a Titan Krios G3i microscope at University of Tokyo (Thermo Fisher Scientific, Rockford, IL) at 300 keV equipped with a VPP, a Gatan Quantum-LS Energy Filter (Gatan, Pleasanton, CA) with a slit width of 20 eV, and a Gatan K3 Summit direct electron detector in the electron counting mode. The nominal magnification was set to ×35,000 with a physical pixel size of 2.67 Å/pixel. Movies were acquired using the SerialEM software version 3.7[45] and the target defocus was set to 1–2 μm. The angular range of the tilt series was from −60° to 60° with 3.0° increments. Each movie was recorded for 1.5 s with a total dose of 2.5 electrons/Å$^2$ and subdivided into 20 frames. The total dose for one tilt series acquisition is thus 100 electrons/Å$^2$. The VPP was advanced to a new position every 1–2 tilt series. When moved to a new position, the VPP was irradiated for 30–60 s by illuminating a blank area before proceeding to the next tilt series acquisition.

**Data processing**. Movies were subjected to beam-induced motion correction using MotionCor2[37], and tilt series images were aligned, CTF corrected, and back-projected to reconstruct 3D tomograms using the IMOD software package version 4.9.12[46]. The estimated defocus values were within the range of 1.1–2.0 μm. Tomograms were 2 × binned (pixel size of 5.34 Å) to reduce the loads of the calculation. Cross-sections of the Z-discs were displayed using the slicer option of 3dmod and the lattice points of the F-actins were manually picked to define the centers of the subtomograms. Volumes with 50 × 50 × 50 pixel-dimensions were cut out from 8 × binned tomograms and were averaged using the PEET software suite version 1.14.0[47]. Averaged subtomograms of thin-section tomography were used for the initial reference, and the alignment was repeated three times for 8 × binned, twice for 4 × binned, once for 2 × binned tomograms. The coordinates and the subtomograms with 200 × 200 × 200 pixel-dimensions (5.34 Å/pixel) were imported to Relion-3 [38] and further refined using conventional 3D refinement and 3D multi-body refinement[21].

The essence of the multibody refinement is performing particle subtractions and local refinements for each flexible domain. Taking a two-body refinement of F-actin and actinin for example, you have an actinin bound to a F-actin, but you cannot refine the two structures together because the actinin is highly flexible relative to the F-actin. In the multibody refinement, you first need to perform a conventional 3D refinement to define the rough positions of the actinin and F-actin. Next, you subtract the actinin densities from subtomograms and align only the F-actin region, and vice versa. In this scheme, you can refine the orientation of each region separately. You repeat the subtraction and refinement steps until convergence. Reconstruction scheme was summarized in Supplementary Fig. 3.

For model building of the α-actinin and the F-actin, the crystal structure of the α-actinin (PDBID: 4D1E)[3] and the F-actin model (PDBID: 6KLL)[48] were fitted to the refined maps using UCSF Chimera[49] only by translations and rotations, without any changes in the coordinates of the models. The morphing movie was generated using UCSF Chimera X[50].

**Statistics and reproducibility**. The numbers of tomograms and subtomograms used for the final reconstructions were as follows; EGTA + ATP state: 43 tomograms and 8,854 subtomograms; Ca + ATP state: 43 tomograms, and 12,535 subtomograms (Table 1). Resolutions of the maps were determined using the gold standard Fourier shell correlation plots (Supplementary Fig. 1c).

**Reporting summary**. Further information on research design is available in the Nature Research Reporting Summary linked to this article.

## Data availability
The averaged maps will be available at the EM Data Bank (www.emdatabank.org) upon publication under the following accesstion numbers: EMD-30242 (EGTA + ATP state) and EMD-30243 (Ca + ATP state). Any remaining information can be obtained from the corresponding author upon reasonable request.

## Code availability
Programs used in this study were all open-sourced software.

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

**Table 1 Cryo-EM data collection, refinement, and validation statistics.**

| | EGTA + ATP state (EMD-30242) | Ca + ATP state (EMD-30243) |
|---|---|---|
| Data collection and processing | | |
| Magnification | 33,000 | 33,000 |
| Voltage (kV) | 300 | 300 |
| Electron exposure (e−/Å$^2$) | 60 | 60 |
| Defocus range (μm) | 1.1–2.0 | 1.1–2.0 |
| Pixel size (Å) | 2.67 | 2.67 |
| Symmetry imposed | n/a | n/a |
| Initial particle images (no.) | 12,925 | 16,685 |
| Final particle images (no.) | 8,854 | 12,535 |
| Map resolution (Å) | 23 | 23 |
| FSC threshold | 0.143 | 0.143 |
| Map resolution range (Å) | ~23 | ~23 |

23. Nishizaka, T., Yagi, T., Tanaka, Y. & Ishiwata, S. Right-handed rotation of an actin filament in an in vitro motile system. *Nature* **361**, 269–271 (1993).

24. Burgoyne, T., Morris, E. P. & Luther, P. K. Three-dimensional structure of vertebrate muscle Z-band: the small-square lattice Z-band in rat cardiac muscle. *J. Mol. Biol.* **427**, 3527–3537 (2015).

25. Schroeter, J. P., Bretaudiere, J. P., Sass, R. L. & Goldstein, M. A. Three-dimensional structure of the Z band in a normal mammalian skeletal muscle. *J. Cell. Biol.* **133**, 571–583 (1996).

26. Knappeis, G. G. & Carlsen, F. The ultrastructure of the Z disc in skeletal muscle. *J. Cell. Biol.* **13**, 323–335 (1962).

27. Franzini-Armstrong, C. The structure of a simple Z line. *J. Cell. Biol.* **58**, 630–642 (1973).

28. Yamaguchi, M., Izumimoto, M., Robson, R. M. & Stromer, M. H. Fine structure of wide and narrow vertebrate muscle Z-lines. A proposed model and computer simulation of Z-line architecture. *J. Mol. Biol.* **184**, 621–643 (1985).

29. Rigort, A. & Plitzko, J. M. Cryo-focused-ion-beam applications in structural biology. *Arch. Biochem. Biophys.* **581**, 122–130 (2015).

30. Wagner, F. R. et al. Preparing samples from whole cells using focused-ion-beam milling for cryo-electron tomography. *Nat. Protoc.* **15**, 2041–2070 (2020).

31. Ylänne, J., Scheffzek, K., Young, P. & Saraste, M. Crystal structure of the alpha-actinin rod reveals an extensive torsional twist. *Structure* **9**, 597–604 (2001).

32. Linke, W. A., Popov, V. I. & Pollack, G. H. Passive and active tension in single cardiac myofibrils. *Biophys. J.* **67**, 782–792 (1994).

33. Stokes, D. L. & DeRosier, D. J. The variable twist of actin and its modulation by actin-binding proteins. *J. Cell. Biol.* **104**, 1005–1017 (1987).

34. Hagen, W. J. H., Wan, W. & Briggs, J. A. G. Implementation of a cryo-electron tomography tilt-scheme optimized for high resolution subtomogram averaging. *J. Struct. Biol.* **197**, 191–198 (2017).

35. Crowther, R. A., Derosier, D. J. & Klug, A. The reconstruction of a three-dimensional structure from projections and its application to electron microscopy. *Proc. Royal Society of London Series. A.* **317**, 319–340 (1970).

36. Danev, R. & Baumeister, W. Cryo-EM single particle analysis with the Volta phase plate. *Elife* **5**, e13046 (2016).

37. Zheng, S. Q. et al. MotionCor2: anisotropic correction of beam-induced motion for improved cryo-electron microscopy. *Nat. Methods* **14**, 331–332 (2017).

38. Zivanov, J. et al. New tools for automated high-resolution cryo-EM structure determination in RELION-3. *Elife* **7**, e42166 (2018).

39. Zhang, P. Advances in cryo-electron tomography and subtomogram averaging and classification. *Curr. Opin. Struct. Biol.* **58**, 249–258 (2019).

40. Li, Z. et al. Subnanometer structures of HIV-1 envelope trimers on aldrithiol-2-inactivated virus particles. *Nat. Struct. Mol. Biol.* **27**, 726–734 (2020)

41. Bui, K. H. & Ishikawa, T. 3D structural analysis of flagella/cilia by cryo-electron tomography. *Methods Enzymol.* **524**, 305–323 (2013).

42. Oda, T. Three-dimensional structural labeling microscopy of cilia and flagella. *Microscopy (Oxf)* **66**, 234–244 (2017).

43. McDonald, K. L. & Auer, M. High-pressure freezing, cellular tomography, and structural cell biology. *Biotechniques* **41**, 137–139 (2006).

44. Oda, T., Abe, T., Yanagisawa, H. & Kikkawa, M. Structure and function of outer dynein arm intermediate and light chain complex. *Mol. Biol. Cell.* **27**, 1051–1059 (2016).

45. Mastronarde, D. N. Automated electron microscope tomography using robust prediction of specimen movements. *J. Struct. Biol.* **152**, 36–51 (2005).

46. Kremer, J. R., Mastronarde, D. N. & McIntosh, J. R. Computer visualization of three-dimensional image data using IMOD. *J. Struct. Biol.* **116**, 71–76 (1996).

47. Nicastro, D. et al. The molecular architecture of axonemes revealed by cryoelectron tomography. *Science* **313**, 944–948 (2006).

48. Oda, T., Yanagisawa, H. & Wakabayashi, T. Cryo-EM structures of cardiac thin filaments reveal the 3D architecture of troponin. *J. Struct. Biol.* **209**, 107450 (2020).

49. Pettersen, E. F. et al. UCSF Chimera–a visualization system for exploratory research and analysis. *J. Comput. Chem.* **25**, 1605–1612 (2004).

50. Goddard, T. D. et al. UCSF ChimeraX: Meeting modern challenges in visualization and analysis. *Protein Sci.* **27**, 14–25 (2018).

## Acknowledgements

This research is partially supported by Platform Project for Supporting Drug Discovery and Life Science Research (Basis for Supporting Innovative Drug Discovery and Life Science Research (BINDS)) from Japan Agency for Medical Research and Development (AMED) under Grant Number JP19am0101115. Computational resource of SGI Rackable C1102-GP8 (Reedbush-H/L) was provided by Initiative on Promotion of Supercomputing for Young or Women Researchers, Supercomputing Division, Information Technology Center, the University of Tokyo. This work was supported by the Takeda Science Foundation (to T.O.) and the Naito Foundation (to T.O.). Molecular graphics and analyses performed with UCSF ChimeraX, developed by the Resource for Biocomputing, Visualization, and Informatics at the University of California, San Francisco, with support from National Institutes of Health R01-GM129325 and the Office of Cyber Infrastructure and Computational Biology, National Institute of Allergy and Infectious Diseases.

## Author contributions

T.O. designed the research; T.O. and H.Y. analyzed data and wrote paper.

## Competing interests

The authors declare no competing interests.
