## [Peer Review File · Communications Biology]

Reviewers' comments:

Reviewer #1 (Remarks to the Author):

This Manuscript presents the long-awaited native structure of mammalian Z-disk derived by cryo-electron tomography of cardiac myofibrils. The structural information of Z-discs have been elusive for decades until the avenues of the "revolution resolution" made this finally possible. The study was performed on isolated myofibrils in the presence of EGTA+ATP and calcium+ATP, representing relaxed or activated states, respectively. The study unravels actin in the basket-weave lattice for the EGTA form and in the diamond-shaped lattice for the Ca+ATP form. The authors propose that the diamond-shaped form, which is thought to be under the mechanical stress due to sliding between thin and thick filaments, converts to the low energy relaxed state through swinging motion of the α -actinin and its elastic properties. This study is definitely an important milestone in understanding structure and ultra-structure of mammalian muscle Z-disks. There are nevertheless a few issues that the author would need to tackle prior to accepting the manuscript for publication:

- The resolution of the generated electron density maps is not explicitly stated anywhere in the text, there is only a reference to the Figure S1 (page 7) where Fourier shell correlation plots are reported for averaged tomograms and for the maps after multibody refinements. I think it is important to spell the resolution out in the text and compare it explicitly to the recently published reconstructions by Burgoyne et al., 2019 and by Rusu et al., 2017, which are also cited in the manuscript.
- Crystal structure of α -actinin-2 dimer was used for fitting to the derived electron density maps, since "resolutions of the α -actinin densities were high enough to fit the crystal structure" (page 7). The authors do not comment/explain what changes were applied to the α -actinin structure to fit in the derived electron density. The inspection of the models as well as movie S3, which morphs the conformational changes between diamond and weave-basketed forms shows several unexpected potentially not plausible features:
 - The α -actinin dimer is stabilised by a series of interactions between adjacent spectrin repeats (SF) of antiparallel subunits; in the F-actin: α -actinin structure, in particular, the EGTA-ATP and CA+ATP structures, the interactions between SR1 and SR4 seem to be lost. This is very unusual for the very stable quaternary structure of α -actinin
 - Actin binding domains of spectrin family of proteins were suggested to undergo a conformational change, i.e. opening of the tandem calponin homology (CH) domains, as part of the regulatory mechanism of the interaction mechanism and reduction of the steric clash between a closed ABD conformation and F-actin. The inspection of the F-actin:EGTA+ATP models displays some clashes (e.g. between chains e and 5), and on the other very loose, or better no real contact between α -actinin and F-actin (e.g. chain h). How was ABD docked to F-actin, and why are there notable differences?
 - As explained by the authors, F-actin:CA+ATP model shows on the other had predominantly very "loose" contacts, which raises a few questions:
 - Does this interaction resemble any known observed F-actin:ABD interactions, and how does it agree with the known F-actin binding sites on ABD and ABD binding sites on F-actin. Recent publication on F-actin decorated by ABD shall be checked as well.
 - F-actin:ABD binding is based on electrostatic interactions, which can be masked by increasing the ionic strength of the solution. For this study 2 mM CaCl₂ was used for treating thin myofibrils; concentration of calcium concentration would be around 100 nM for relaxation and at 5 μ M for contraction in skeletal muscle, which is much lower than the concentrations used here. How can the authors exclude that the ABD detachment was not at least in part due to electrostatic shielding coming from 2 mM CaCl₂?
 - In the presented F-actin: α -actinin the EF3-4 hands are not bound to the neck region connecting ABD and SR1, which is expected since they shall be engaged in interaction with titin. Once EF-34 detaches from the neck, the latter is expected to unfold, to allow for the required

positioning/reorientation of ABD suitable for interaction with F-actin. The authors might like to attend to this in their models. How was EF1-2 modelled/positioned?

Reviewer #2 and #4 (Remarks to the Author):

Cryo-electron tomography of cardiac myofibrils reveals a 3D lattice spring within the Z-discs

Toshiyuki Oda, and Haruaki Yanagisawa

This study presents excellent results on the fine structure of the Z-disc in cardiac muscle and how the Z-disc responds to change of state from relaxed to active muscle. The authors have to be congratulated on their achievement. However, in its current form this paper is substantially lacking in the way in which these new results are discussed and interpreted in particular with respect to previous studies. It is our opinion that a major revision in this area is required in order to make the paper suitable for publication in *Communications Biology*.

Comments:-

1) The basketweave, small-square and diamond lattice forms of the Z-disc. In their introduction the authors discuss the two distinct and well known structures previously described for the Z-disc: the small-square and basketweave forms which can be identified in the Z-disc when viewed transversely. They point out that their structure of the cardiac muscle Z-disc prepared in relaxing solution is highly compatible with previously described structures of the basketweave form of the Z-disc, albeit at greatly increased resolution. They show that the characteristic basketweave appearance arises from curved alpha-actinin dimers running between actin filaments of opposite polarity. A similar interpretation had been proposed previously by Burgoyne et al (2019). However, the substantially improved resolution in the current paper allows the direct identification of the alpha-actin dimers and their mode of interaction with the actin filaments with great confidence.

In contrast, their Z-disc structure obtained from cardiac myofibrils prepared in an activating solution showed straight alpha-actinin dimers running between actin filaments of opposite polarity. The authors compare this with the small-square Z-disc, in particular the structure of Burgoyne et al (2015) and consider that they both share a straight conformation of alpha-actinin. However, although the small square Z-disc as described by Burgoyne et al (2015) and a number of previous studies is indeed characterised in transverse view by straight connecting densities that appear to run between actin filaments, the apparent connection is between actin filaments of the same polarity. Therefore, it seems that the new structure presented here for the active state is fundamentally different from the small-square form. Instead it seems to be rather similar to the diagonal square net described by Yamaguchi et al (1985) *JMB* 184: 621-644 which has links running directly between actin filaments of opposite polarity.

2) Section studies. Since the authors have used thin sections cut from resin-embedded tissue, they should show some example electron micrographs of the EGTA and Ca samples, maybe in the Supplementary Information. Readers would also be interested to see low-magnification views of the cardiac myofibrils. The authors should clarify whether they used longitudinal or transverse sections for the tomography. The authors used the averaged tomograms from the thin sections as references for the cryo-em analysis. It is well known that resin sections undergo shrinkage in depth as well as in the plane of the section. How did the authors compensate for the shrinkage?

3) The Z-disc structure under activating conditions. The authors investigated the active state of the Z-disc by incubation of cardiac myofibrils with Ca²⁺ and ATP. They state "We compared the sarcomere lengths in the EGTA+ATP and the Ca+ATP states and confirmed that the Ca+ATP treatment properly activated the myofibrils (ter Keurs et al., 1980) (Fig. S1A)". However, changes

in sarcomere length are not necessarily an effective way to assess complete activation in myofibrils. The study by ter Keurs relates to trabecula muscle which are tethered at both ends. The authors' myofibrils are free, not tethered. So for the Ca sample, they could confirm that the muscle was in active state by following the method of Luther & Squire, JMB, 2002,319:1157-1164, see Fig 3, who looked at longitudinal sections which include the A-band and Z-disc.

Furthermore, in their section headed "Mechanism of the conformational change", the authors consider that the conformation of their activated state may be driven by mechanical stress. It is not clear in their experiment with untethered myofibrils whether there would be appreciable mechanical stress in the system. Do the authors have evidence for such stress?

Heterogeneity. The authors see quite a lot of heterogeneity in the analysis of their subtomograms which they discuss in terms of actin filaments having an angular rise of 166.6 which gives a ~86 degrees rotation every 7 subunits. However, they seem to ignore the well-established tendency of the angular rise to vary in actin filaments. So in fact exact 90 degree rotations every 7 subunits is quite possible which would remove any requirement for intrinsic heterogeneity in the system.

4) Additional proteins. The authors' Z-disc structures have been very effectively modelled with atomic models of actin and alpha-actinin. However, the whole Z-disc is composed of a plethora of proteins in addition to actin and alpha-actinin, eg titin and nebulin (see Luther, J.MuscleRes. CellMotil. 30 (2009) 171-185). Unfortunately, the authors do not identify any additional proteins in their structures. Especially interesting would be the interaction between the alpha-actinin C-terminal Cam-like domain and the Zr repeats of titin. Maybe the authors could investigate if there is any sign of the complex Act-EF34-Zr7 (PDB 1H8B) as reported by Atkinson et al (2001) Nat Str Biol, 8:853. Additionally, the authors should try to fit CapZ into the barbed-end of the actin filaments.

5) Line 128: "it is likely that the interface between the F-actin and the actin-binding domain of the alpha-actinin is not strictly defined, which is similar to the F-actin tropomyosin association". This direct comparison needs to be better explained. The F-actin tropomyosin association and relative movement is mediated by the troponin complex and involves the simultaneous interaction between seven distinct interfaces on each tropomyosin molecule. Accordingly, the energy associated with the interaction of tropomyosin with each individual actin will be rather low. In contrast, alpha-actinin appears to form a single interface with each actin filament with which it interacts.

Reviewer #3 (Remarks to the Author):

Summary: This paper describes a new reconstruction obtained from cryoelectron tomography of the Z-disk from vertebrate cardiac muscle. The authors use a number of technical advances to obtain what is at face value the best 3-D image of a vertebrate Z-disk yet obtained. In fact, it is so much better than anything previously reported that it should be considered ground breaking.

Major Comments:

Aside from the improved equipment used for both the specimen preparation and the data collection, the authors have improved the reconstruction by averaging thin filaments and cross linkers separately from unit cells, which has enabled them to leapfrog the primary issue that has limited Luther's work on the vertebrate Z-disk, which utilized spatial averaging and thus could not treat heterogeneity adequately. I would applaud them on that.

The data acquisition and preprocessing appear to be 1st class. The authors used an approach that has become popular when a tomogram is well populated by equivalent or quasiequivalent

structures. Recognizing that the specimen thickness was appreciably too great for obtaining much detail from subtomogram averaging, they used the subtomogram averaging to obtain coordinates for the basic unit, an antiparallel pair of actin filaments and the intervening alpha-actinin and used multibody reconstruction to obtain the result presented, which exceeds the resolution previously reported by anyone by at least a factor of 2.

For specimens of thickness 200-300 nm thick tilted at increments of 3° the Crowther criterion formulated for a slab specimen, the limiting resolution would be between 105 and 157 Å

$$\begin{aligned} \text{resolution} &= \text{thickness} * \tan(3^\circ) = 2000 \text{ \AA} * 0.0524 = 105 \text{ \AA} \\ &= 3000 \text{ \AA} * 0.0524 = 157 \text{ \AA} \end{aligned}$$

I think it is probably worth mentioning how limiting the subtomogram averaging approach by itself would be to help the non-specialist reader to appreciate just how much their present result improves the Z-disk image.

If Communications Biology is targeting a wide audience, not just cryoEM afficiandos or Z-disk fanatics, it might be worth a few sentences to describe how multibody refinement circumvents the limits of subtomogram averaging.

The movies have pitifully little detail in their legends. The movies themselves are high quality. I believe the movie legends should be more complete, even at the risk of repeating information in the static figure legend. However, these were made to show the authors interpretation of the data and must have come from subtomograms reassembled from averages. I believe it is essential to provide movies equivalent in view to those that show the subtomogram averages, on which the interpretation movies are based. This adds two new movies, which I think is not too much to ask.

The interpretation of the work as illustrated in Movie 3 raises some interesting questions. The movie itself I would complement them on as it is quite clarifying. Not only do the filaments move, but they also rotate about their axis. For the filaments to rotate about their axis, some entity must be applying a torque. What is that entity? If a torque is applied, should this not manifest itself in a change in helical angle of the thin filaments. However, there is no evidence that anything other than the axial spacing between actin subunits changes when the filament is activated; no one has observed a change in the spacing of the genetic helices, which if there is a change in pitch would alter their spacing. There is evidence in the literature on myosin motors applying a torque to the filaments in motility assays suggesting that the filaments rotate about their helical axis as if they were rigid bodies. In addition, the filaments move laterally, which is not so surprising since the square lattice must deform into the hexagonal lattice of the A-band. This would be the first reconstruction to provide a detailed model for how this happens.

Minor comments:

II. 50-51. "...the variable ionic environment within the cytoplasm of the muscle ..." Exactly which ionic constituent that an experimenter could control varies in concentration within the muscle cytoplasm? Perhaps a citation might be useful here.

II. 60-61. "...Although the Z-discs in the relaxed (EGTA+ATP) state showed a square lattice as expected (Fig. 1A and C), ..." Ambiguous. Both the basket weave and the small square forms show a square lattice. Should be clarified.

II. 78-79 "We extracted subtomograms based on the lattice points appeared on the cross-sections ..." This wording is quite awkward as I cannot determine what was meant. Consider revising.

I. 85. "conducted" A seldom used word, though correctly used in this case. Causes the reader to look it up, which breaks up their reading. Consider changing.

ll. 106-107. Because the eigenvectors describe molecular motions, I believe that one or more movies are warranted since a static image does not adequately describe motion.

ll 142-150: Have the authors considered that the deformed angles provided in this paragraph, which all come from sectioned muscle, might differ from 90° due to section compression? In fact I believe that Luther recently published a Z-disk structure in which he specifically corrected for these distortions due to section compression.

line 266: What values of CTF were obtained from the CTF determination? Did they fall within the selected range? Provide the range of determined values.

ll. 488-489. Given the thickness of the specimen and the tilt increment, I think that a value of 31-32 Å is overly optimistic and simply illustrates the inadequacy of the FSC as a measure of resolution. Since it is not critical to their result, I would leave it out.

Figure S1. I find it interesting that the myofibrils, which are unconstrained, only shortened a limited amount in calcium+ATP. Had they been skeletal muscle myofibrils, they would hypershortened and destroyed the structure. To what do the authors attribute the fact that the myofibrils only shortened a small amount when activated.

Figure 5. The conformational change in alpha-actinin is displayed as a change in F-actin binding. To show the change in the alpha-actinin, it would be better to show the two structures displayed overlayed on each other without the F-actin. Since they are talking about conformational changes, a movie morphing between the two would be quite informative.

Reviewers' comments and **Authors' replies**:

Reviewer #1 (Remarks to the Author):

This Manuscript presents the long-awaited native structure of mammalian Z-disk derived by cryo-electron tomography of cardiac myofibrils. The structural information of Z-discs have been elusive for decades until the avenues of the “revolution resolution“ made this finally possible. The study was performed on isolated myofibrils in the presence of EGTA+ATP and calcium+ATP, representing relaxed or activated states, respectively. The study unravels actin in the basket-weave lattice for the EGTA form and in the diamond-shaped lattice for the Ca+ATP form. The authors propose that the diamond-shaped form, which is thought to be under the mechanical stress due to sliding between thin and thick filaments, converts to the low energy relaxed state through swinging motion of the α -actinin and its elastic properties.

This study is definitely an important milestone in understanding structure and ultra-structure of mammalian muscle Z-disks. There are nevertheless a few issues that the author would need to tackle prior to accepting the manuscript for publication:

The resolution of the generated electron density maps is not explicitly stated anywhere in the text, there is only a reference to the Figure S1 (page 7) where Fourier shell correlation plots are reported for averaged tomograms and for the maps after multibody refinements. I think it is important to spell the resolution out in the text and compare it explicitly to the recently published reconstructions by Burgoyne et al., 2019 and by Rusu et al., 2017, which are also cited in the manuscript.

Authors Reply1: We explicitly mentioned the resolution of the refined maps and compared it with the previous reports as below:

Page7, line 96: *“We repeated this multibody refinement for all the four anti-parallel F-actin pairs, and reconstructed the whole repeat unit of the Z-disc with $\sim 23 \text{ \AA}$ (Fig. S1C, Fig. 4A, and Movies S2), which was significantly improved compared to the previous results (39 \AA^{14} and 58 \AA^{22}).”*

- Crystal structure of α -actinin-2 dimer was used for fitting to the derived electron density maps, since “resolutions of the α -actinin densities were high enough to fit the crystal structure” (page 7). The authors do not comment/explain what changes were applied to the α -actinin structure to fit in the derived electron density.

Authors' reply: Because the resolutions of our maps were not high enough for flexible docking, we fitted the modes directly into our maps without changing the coordinates of the models. Thus, we weakened the statement as below:

Page 7, line 99: “*The resolutions of the α -actinin densities were sufficient for rigid-body docking of the crystal structure*”

The inspection of the models as well as movie S3, which morphs the conformational changes between diamond and weave-basket forms shows several unexpected potentially not plausible features:

- The α -actinin dimer is stabilized by a series of interactions between adjacent spectrin repeats (SF) of antiparallel subunits; in the F-actin: α -actinin structure, in particular, the EGTA-ATP and CA+ATP structures, the interactions between SR1 and SR4 seem to be lost. This is very unusual for the very stable quaternary structure of α -actinin (Ylanne, J., et al Structure 2001).

Authors' reply: Thank you for pointing out the essential point of our model interpretation. Because the resolutions of our maps were limited to $\sim 23 \text{ \AA}$, the boundary between the two monomers of the actinin was unclear. As a result of masking function of the multibody refinement algorithm, the actinin dimer was forced to be divided into two parts, which does not necessarily a correct division. In Supplemental Movie 3, you can see the undivided maps of actinin dimer. Although we fitted the crystal structure of actinin without changing the coordinates, it would be necessary to consider the rotation around the SR domains relative to the ABD. However, flexible fitting requires at least 8 \AA resolution. Thus, we would leave the accurate fitting of the actinin models for future studies. This reply continues to the next question.

- Actin binding domains of spectrin family of proteins were suggested to undergo a conformational change, i.e. opening of the tandem calponin homology (CH) domains, as part of the regulatory mechanism of the interaction mechanism and reduction of the steric clash between a closed ABD confirmation and F-actin. The inspection of the Fa-actin:EGTA+ATP models displays some clashes (e.g. between chains e and 5), and on the other very loose, or better no real contact between α -actinin and F-actin (e.g. chain h). How was ABD docked to F-actin, and why are there notable differences?

Authors' reply: As explained in the previous replies, we docked the actinin crystal structure to our maps only by translation and rotation of the whole model without changing the coordinates. In the multibody refinement, the actinin body and the F-actin body are refined independently from each other. After calculating the ranges of molecular movements (see Supplemental Movie 3), final results are generated by taking the median values in the XYZ translations and the XYZ rotations (Nakane et al *eLife* 2018). Thus, the spatial relationship between the “median” positions of the actinin and the F-actin does not necessarily reflect the actual binding, especially when F-actin shows such broad range of displacements. We added the following paragraph to explain the limitation of our interpretation:

Page 11, line 175: *“Note that we did not perform flexible fitting of the crystal structure to our maps due to the limited the resolution. Therefore, the fitted models were not necessarily accurate, which can be discerned that there are clashes between the EF-hand and the neck region and apparent dissociation between the spectrin repeats (Movie S4-5) ³¹. It is necessary to improve the resolution up to ~8Å to precisely assign the domains of the α -actinin.”*

Comments about the ABD was further addressed in the next reply.

○ As explained by the authors, F-actin:Ca+ATP model shows on the other had predominantly very “loose” contacts, which raises a few questions:

■ Does this interaction resemble any known observed F-actin:ABD interactions, and how does it agree with the known F-actin binding sites on ABD and ABD binding sites on F-actin. Recent publication on F-actin decorated by ABD shall be checked as well. e.g. F-actin decorated by plectin ABD Nat Commun. 2017 Nov 7;8(1):1350. doi: 10.1038/s41467-017-01367-w.

Author's reply: Thank you for suggesting a comparison with recent structural studies of the ABD. In the recent cryo-EM paper of filamin A ABD-F-actin complex (Iwamoto et al Nat. Struct. Mol. Biol. 2018), they achieved 3.5 Å resolution. However, the high-resolution map was derived from an altered protein with a point mutation (E254K). The cryo-EM map of F-actin decorated with the wild-type ABD was low in resolution (9.8 Å), and not only the ABD densities were obscure, but the F-actin densities were also deteriorated (EMD-7833), probably because of heterogeneous binding of the ABD to F-actin. Similarly, the 6.9 Å resolution cryo-EM map of spectrin ABD-F-actin structure was solved with a L235P mutation (Avery et al Nat. Comm. 2017). These reports suggest that the binding between the wild-type ABD and the F-actin is highly heterogeneous. Our docking showed the binding of ABD to the actin subdomain SD2 (Fig. 5B), which basically agrees with the previous reports above. We added the following paragraph to the Discussion section:

Page 11, line 180: *“Although high resolution structures of F-actin decorated with the actin binding domains has been reported ^{32, 33}, the actin binding domains used in these studies have point mutations, which increase the affinity of the domain to the F-actin and probably stabilize the complexes. The structure of F-actin decorated with wild-type actin binding domains is low in resolution and its structural details are blurred, probably due to the structural heterogeneity ³². We suppose that the actin-binding domain of the α -actinin could bind to a wide area of the F-actin surface because the Ca+ATP treatment displaced the α -actinin along the surface of the F-actin (Fig. 5C, right).”*

■ F-actin:ABD binding is based on electrostatic interactions, which can be masked by increasing the ionic

strength of the solution. For this study 2 mM CaCl₂ was used for treating thin myofibrils; concentration of calcium concentration would be around 100 nM for relaxation and at 5 μM for contraction in skeletal muscle, which is much lower than the concentrations used here. How can the authors exclude that the ABD detachment was not at least in part due to electrostatic shielding coming from 2 mM CaCl₂?

Authors' Reply: We added 1 μl of 2 mM CaCl₂ to 4 μl of myofibril suspension, thus the final concentration of CaCl₂ was 0.4 mM. We intentionally used high concentration of CaCl₂ because on-grid mixing can be incomplete and the effective concentration of CaCl₂ at the specimen level can be lower than calculated. Still, the estimated CaCl₂ concentration would be relatively higher than the physiological value, but KCl concentration of our buffer (60 mM) was low enough to compensate the excessive CaCl₂. Therefore, we believe that 0.4 mM CaCl₂ did not significantly affect the electrostatic interaction.

○ In the presented F-actin:α-actinin the EF3-4 hands are not bound to the neck region connecting ABD and SR1, which is expected since they shall be engaged in interaction with titin. Once EF-34 detaches from the neck, the latter is expected to unfold, to allow for the required positioning/reorientation of ABD suitable for interaction with F-actin. The authors might like to attend to this in their models. How was EF1-2 modelled/positioned?

Authors' reply: We docked the whole actinin crystal structure without modifying the coordinates. We can see densities that is likely to correspond to EF-hand part (Supplemental Movie 3), but we could not visualize the unfolding due to the limited resolution. In addition, the neck part is mostly missing in our map. Therefore, we do not think it is meaningful to interpret the details of conformational changes in the EF-hand or spectrin-repeat domains.

- The authors use term “elasticity” of α-actinin; elasticity is in molecular sciences often associated with folding/unfolding, and in particular in muscle field elasticity of titin with its PEVK and Ig-domain folding-unfolding allowing to accommodate the sarcomere contraction/extension during muscle contraction. I think this is not the case here and using the term elasticity might lead to wrong interpretation, I would recommend using expressions like “conformational flexibility or plasticity” or similia.

Authors' reply: We replaced “elasticity” with “flexibility” or “structural plasticity” accordingly.

Reviewer #2 and #4 (Remarks to the Author):

This study presents excellent results on the fine structure of the Z-disc in cardiac muscle and how the Z-disc responds to change of state from relaxed to active muscle. The authors have to be congratulated on their achievement. However, in its current form this paper is substantially lacking in the way in which these

new results are discussed and interpreted in particular with respect to previous studies. It is our opinion that a major revision in this area is required in order to make the paper suitable for publication in Communications Biology.

Comments:-

1) The basketweave, small-square and diamond lattice forms of the Z-disc. In their introduction the authors discuss the two distinct and well known structures previously described for the Z-disc: the small-square and basketweave forms which can be identified in the Z-disc when viewed transversely. They point out that their structure of the cardiac muscle Z-disc prepared in relaxing solution is highly compatible with previously described structures of the basketweave form of the Z-disc, albeit at greatly increased resolution. They show that the characteristic basketweave appearance arises from curved alpha-actinin dimers running between actin filaments of opposite polarity. A similar interpretation had been proposed previously by Burgoyne et al (2019). However, the substantially improved resolution in the current paper allows the direct identification of the alpha-actin dimers and their mode of interaction with the actin filaments with great confidence.

In contrast, their Z-disc structure obtained from cardiac myofibrils prepared in an activating solution showed straight alpha-actinin dimers running between actin filaments of opposite polarity. The authors compare this with the small-square Z-disc, in particular the structure of Burgoyne et al (2015) and consider that they both share a straight conformation of alpha-actinin. However, although the small square Z-disc as described by Burgoyne et al (2015) and a number of previous studies is indeed characterized in transverse view by straight connecting densities that appear to run between actin filaments, the apparent connection is between actin filaments of the same polarity. Therefore, it seems that the new structure presented here for the active state is fundamentally different from the small-square form. Instead it seems to be rather similar to the diagonal square net described by Yamaguchi et al (1985) JMB 184: 621-644 which has links running directly between actin filaments of opposite polarity.

Authors' reply: Thank you for clarifying the issues with the comparison between our observation and the previous works. Due to the limited resolution, the orientation and the binding manner of actinin are not clearly defined in the thin section-based works (Burgoyne et al 2015 and other previous reports) In Fig. 3C of Burgoyne et al JMB 2015, they observed a density that is running between opposite polarity F-actins. Thus, we believe that the structure in Burgoyne et al 2015 is not entirely different from our results. We added the following paragraph to the Discussion:

Page 9, line 152: *“In the previous studies based on the ultra-thin section observations, it has been suggested that the actinin cross-bridges between F-actins of the same polarity under the rigor condition or in the small square form ^{7,9}. In our structures, the actinin cross-bridges were formed between F-actins of opposite polarity in both Ca+ATP and EGTA+ATP states (Fig. 3). In the previous thin section tomography study, however, cross-bridging densities running between F-actins of opposite polarity are observed in the Z-disc in the small square form ²⁴ and a similar model has been proposed as a “diagonal square” form ²⁶⁻²⁸. It is possible that our “diamond-shaped” form corresponds to, or related to the “diagonal square” form, which is thought to be different from the small square form. It would be necessary to conduct a cryo-FIB tomography ^{29,30} of native muscle fibers to make a clear distinction among these forms.”*

2) Section studies. Since the authors have used thin sections cut from resin-embedded tissue, they should show some example electron micrographs of the EGTA and Ca samples, maybe in the Supplementary Information. Readers would also be interested to see low-magnification views of the cardiac myofibrils. The authors should clarify whether they used longitudinal or transverse sections for the tomography. The authors used the averaged tomograms from the thin sections as references for the cryo-em analysis. It is well known that resin sections undergo shrinkage in depth as well as in the plane of the section. How did the authors compensate for the shrinkage?

Authors' reply: We added low-magnification images of the cardiac myofibrils (Fig. S1A). We used the transverse views for generating the reference volume.

Although we were aware that there was a shrinkage problem, we did not modify the thin section map because it was used only for the first iteration in the alignment of the subtomograms. After the first iteration, the thin section-derived reference was discarded and was replaced with cryo EM-derived initial average. We added the following sentence to the Method section:

Page 17, line 293: *“Although the epon-embedded specimens were likely to be deformed by shrinkage of the section ¹⁴, we believe that it did not affect the averaging of the cryo tomograms because the section-derived maps were used only in the first iteration and subsequently replaced with newly generated averages of cryo tomograms.”*

3) The Z-disc structure under activating conditions. The authors investigated the active state of the Z-disc by incubation of cardiac myofibrils with Ca²⁺ and ATP. They state “We compared the sarcomere lengths in the EGTA+ATP and the Ca+ATP states and confirmed that the Ca+ATP treatment properly activated the myofibrils (ter Keurs et al., 1980) (Fig. S1A)”. However, changes in sarcomere length are not necessarily an effective way to assess complete activation in myofibrils. The study by ter Keurs relates to trabecula muscle which are tethered at both ends. The authors' myofibrils are free, not tethered. So for the

Ca sample, they could confirm that the muscle was in active state by following the method of Luther & Squire, JMB, 2002,319:1157-1164, see Fig 3, who looked at longitudinal sections which include the A-band and Z-disc.

Reviewer's additional comment.

I am afraid that I have made an error here and my suggested method to test for activation is invalid. Nevertheless Dr XXX and I are concerned about the activation state of their myofibrils and whether they are fully activated in their unrestrained state. Certainly their reference, ter Keurs et al., 1980, is invalid as ter Keurs used tethered muscle not unrestrained myofibrils as in the authors' case.

Authors' reply: Thank you for pointing out an essential issue about the mechanical state of myofibrils. We agree that the myofibrils suspended in the buffer were definitely not under tension. However, when we applied the myofibrils to the holey carbon grid and blotted the buffer from the back-side, the myofibrils were pressed against the carbon membrane, while part of the myofibrils was suspended over the hole. Therefore, it is reasonable to assume that the myofibrils were tethered to the carbon membrane and the recorded regions of the myofibrils suspended over the holes could be under tension, in a situation similar to the conventional force measurement of myofibrils. It is difficult to confirm that the myofibrils were actually under tension, but we examined the state of the myosin motors by averaging the thin filaments in the A-band (Fig. S2). The changes in the association of myosin heads to the thin filaments partly supports our conclusion that the Ca+ATP myofibrils were in the activated state. We added a section to Discussion to address this issue as below:

Page 12, line 193: "*Mechanical state of the myofibrils*

It is uncertain if the hydro-frozen myofibrils observed in this study were actually under tension because the myofibrils were extracted from cells and were not anchored to glass needles, which are conventionally used for force measurements³⁶. However, we believe that the myofibrils were supposed to be under tension at the moment of pluge-freezing because most part of these fibers were pressed against the carbon membrane, while the imaged regions were suspended within the water film encompassing the holes in the carbon membrane (Fig. 2). Measurement of the sarcomere length partly supported this conclusion (Fig. S1B). We further examined the state of the myofibrils by averaging the thin filaments in the A-band (Fig. S2). In the EGTA+ATP state, the myosin heads were completely dissociated from the thin filament. Both in the Ca-only and the Ca+ATP states, by contrast, the myosin heads fully decorate the thin filament. In the Ca+ATP state, however, the structure was partially disordered, especially at the F-actin region, probably due to the active power strokes of the myosin and resulting heterogenous spatial relationship between the myosin head and the F-actin. These observations support that the myofibrils were properly activated by addition of Ca and ATP."

We also added a description of the back-side blotting in the Method section:

Page 18, line 306: *“We applied a small offset to the positions of the blotting arms so that one of the filter papers reached the grid slightly earlier than the other, and thus most of the buffer was blotted from the back-side of the grid. This back-side blotting significantly increased the filament density on the grid.”*

Furthermore, in their section headed “Mechanism of the conformational change”, the authors consider that the conformation of their activated state may be driven by mechanical stress. It is not clear in their experiment with untethered myofibrils whether there would be appreciable mechanical stress in the system. Do the authors have evidence for such stress?

Authors' reply: As explained in the previous question, we believe that myofibrils were under tension because myofibrils were absorbed onto the carbon membrane. The sarcomere length measurement (Fig. S1B) and the thin section structures (Fig. S2) suggest activation of the myofibrils.

Heterogeneity. The authors see quite a lot of heterogeneity in the analysis of their subtomograms which they discuss in terms of actin filaments having an angular rise of 166.6 which gives a ~86 degrees rotation every 7 subunits. However, they seem to ignore the well-established tendency of the angular rise to vary in actin filaments. So in fact exact 90 degree rotations every 7 subunits is quite possible which would remove any requirement for intrinsic heterogeneity in the system.

Author's reply: Thank you for pointing out an alternative model. We mentioned the possibility as below:

Page 13, line 223: *“Note that the basket-weave square lattice with 90°/90° rotation angles is considered to be structurally favorable over the diamond-shaped lattice because the basket-weave form has been observed in relaxed/unstrained state. The variation in the helical twist angles of F-actin³⁷ could compensate the discrepancy between the square lattice and the helical symmetry of F-actin.”*

4) Additional proteins. The authors' Z-disc structures have been very effectively modelled with atomic models of actin and alpha-actinin. However, the whole Z-disc is composed of a plethora of proteins in addition to actin and alpha-actinin, eg titin and nebulin (see Luther, J.MuscleRes. CellMotil. 30 (2009) 171-185). Unfortunately, the authors do not identify any additional proteins in their structures. Especially interesting would be the interaction between the alpha-actinin C-terminal Cam-like domain and the Zr repeats of titin. Maybe the authors could investigate if there is any sign of the complex Act-EF34-Zr7 (PDB 1H8B) as reported by Atkinson et al (2001) Nat Str Biol, 8:853. Additionally, the authors should try to fit CapZ into the barbed-end of the actin filaments.

Authors' reply: Thank you for interesting suggestions. We attempted to visualize other components of the Z-disc by performing 3D classification. However, classification was unsuccessful because the missing wedge imposed a strong bias on the classification, and because the distance between the center of the subtomogram and the barbed-end of the F-actin was highly variable. For visualization of other components, it would be necessary to label the molecules using antibodies or protein-tags to emphasize their localization.

5) Line 128: "it is likely that the interface between the F-actin and the actin-binding domain of the alpha-actinin is not strictly defined, which is similar to the F-actin tropomyosin association". This direct comparison needs to be better explained. The F-actin tropomyosin association and relative movement is mediated by the troponin complex and involves the simultaneous interaction between seven distinct interfaces on each tropomyosin molecule. Accordingly, the energy associated with the interaction of tropomyosin with each individual actin will be rather low. In contrast, alpha-actinin appears to form a single interface with each actin filament with which it interacts.

Author's reply: We found that referring to tropomyosin was not appropriate in this context. Thus, we removed the sentence.

Reviewer #3 (Remarks to the Author):

Summary: This paper describes a new reconstruction obtained from cryoelectron tomography of the Z-disk from vertebrate cardiac muscle. The authors use a number of technical advances to obtain what is at face value the best 3-D image of a vertebrate Z-disk yet obtained. In fact, it is so much better than anything previously reported that it should be considered ground breaking.

Major Comments:

For specimens of thickness 200-300 nm thick tilted at increments of 3° the Crowther criterion formulated for a slab specimen, the limiting resolution would be between 105 and 157 Å

$$\begin{aligned} \text{resolution} &= \text{thickness} * \tan(3^\circ) = 2000 \text{ \AA} * 0.0524 = 105 \text{ \AA} \\ &= 3000 \text{ \AA} * 0.0524 = 157 \text{ \AA} \end{aligned}$$

I think it is probably worth mentioning how limiting the subtomogram averaging approach by itself would be

to help the non-specialist reader to appreciate just how much their present result improves the Z-disk image.

Authors' reply: Thank you for the valuable suggestion. We added the following sentences to explain the relationship between the angular increment of the tilt series and the resolution.

Page 14, line 235: *"The highest resolution achieved in this study was 23 Å, which is insufficient to conduct a flexible docking of crystal structure. Cryo-electron tomography of thick specimen is challenging due to several factors that affect attainable resolution³⁸. According to the Crowther criterion $d = \pi * D/m$ (resolution, d ; particle diameter, D ; and number of images, m)³⁹, the resolution limit of a single tomogram is estimated to be 157 Å. In the case of Z-disc tomography, we could overcome this limit because the F-actin lattices were randomly oriented relative to the beam axis."*

If Communications Biology is targeting a wide audience, not just cryoEM aficionados or Z-disk fanatics, it might be worth a few sentences to describe how multibody refinement circumvents the limits of subtomogram averaging.

Authors' reply: We added a brief description of the multibody refinement as below:

Page 19, line 339: *"The essence of the multibody refinement is performing particle subtractions and local refinements for each flexible domain²¹. Taking a two-body refinement of F-actin and actinin for example, you have an actinin bound to a F-actin, but you cannot refine the two structures together because the actinin is highly flexible relative to the F-actin. In the multibody refinement, you first need to perform a conventional 3D refinement to define the rough positions of the actinin and F-actin. Next, you subtract the actinin densities from subtomograms and align only the F-actin region, and vice versa. In this scheme, you can refine the orientation of each region separately. You repeat the subtraction and refinement steps until convergence."*

The movies have pitifully little detail in their legends. The movies themselves are high quality. I believe the movie legends should be more complete, even at the risk of repeating information in the static figure legend. However, these were made to show the authors interpretation of the data and must have come from subtomograms reassembled from averages. I believe it is essential to provide movies equivalent in view to those that show the subtomogram averages, on which the interpretation movies are based. This adds two new movies, which I think is not too much to ask.

Authors' reply: We added the following supplemental movies:

Movie S1: Averaged tomograms in the EGTA+ATP and the Ca+ATP states.

Movie S5: Morphing movie of actinin models only.

We added descriptions of the maps and models to the legends of the supplemental movies as below:

“Supplemental movie 2

Movie representation of Fig. 4A. Composite maps in the EGTA+ATP and the Ca+ATP states, composed of one central F-actin (gray), four opposite-polarity F-actins (yellow), sixteen α -actinin monomers (orange and green). The α -actinin crystal structures (PDB 4D1E) were fitted into the α -actinin maps (red and green models).”

The interpretation of the work as illustrated in Movie 3 raises some interesting questions. The movie itself I would complement them on as it is quite clarifying. Not only do the filaments move, but they also rotate about their axis. For the filaments to rotate about their axis, some entity must be applying a torque. What is that entity? If a torque is applied, should this not manifest itself in a change in helical angle of the thin filaments. However, there is no evidence that anything other than the axial spacing between actin subunits changes when the filament is activated; no one has observed a change in the spacing of the genetic helices, which if there is a change in pitch would alter their spacing. There is evidence in the literature on myosin motors applying a torque to the filaments in motility assays suggesting that the filaments rotate about their helical axis as if they were rigid bodies. In addition, the filaments move laterally, which is not so surprising since the square lattice must deform into the hexagonal lattice of the A-band. This would be the first reconstruction to provide a detailed model for how this happens.

Authors' reply: Thank you for pointing out a possible torque applied by myosin motors. We added the following sentences:

Page 8, line 113: *“It has been reported that myosin motors apply a right-handed torque to the F-actin²³. The axial rotation observed in our maps is likely to be related to this myosin-dependent screw rotation.”*

Minor comments:

II. 50-51. “...the variable ionic environment within the cytoplasm of the muscle ...” Exactly which ionic constituent that an experimenter could control varies in concentration within the muscle cytoplasm?

Perhaps a citation might be useful here.

Authors' reply: As below: we cited Marcucci et al 2018, which demonstrates variation of calcium concentration within the cytoplasm of muscle fiber due to diffusion and micro-domain-compartmentalization.

Page 3, line 46: *"the variable ionic environment within the cytoplasm of the muscle tissue specimen makes it difficult to obtain a regular lattice structure of the Z-discs ¹⁵."*

II. 60-61. "...Although the Z-discs in the relaxed (EGTA+ATP) state showed a square lattice as expected (Fig. 1A and C), ..." Ambiguous. Both the basket weave and the small square forms show a square lattice. Should be clarified.

Authors' reply: We corrected the sentences as below:

Page 5, line 58: *"Although the Z-discs in the relaxed (EGTA+ATP) state showed a square lattice with inter-actin connections"*

II. 78-79 "We extracted subtomograms based on the lattice points appeared on the cross-sections ..." This wording is quite awkward as I cannot determine what was meant. Consider revising.

Authors' reply: We corrected the sentence as below:

Page 6, line 76: *"Lattice points of the F-actin were clearly observed in the cross-sections of the Z-discs (Fig. 2E). We extracted subtomograms using these lattice points as centers."*

I. 85. "conduced" A seldom used word, though correctly used in this case. Causes the reader to look it up, which breaks up their reading. Consider changing.

Authors' reply: We corrected "conduced" to "conducted".

II. 106-107. Because the eigenvectors describe molecular motions, I believe that one or more movies are warranted since a static image does not adequately describe motion.

Authors' reply: we added the supplemental movie 3 to show the eigenvector motions.

II 142-150: Have the authors considered that the deformed angles provided in this paragraph, which all come from sectioned muscle, might differ from 90° due to section compression? In fact I believe that Luther recently published a Z-disk structure in which he specifically corrected for these distortions due to section compression.

Authors' reply: Thank you for pointing out the possibility of compression. It is possible that the thin section-derived maps were deformed by compression, but we used these maps only in the first iteration of the subtomogram alignments. After the first alignment, these maps were replaced by cryoEM-derived new averages. Thus, the possible effect of compression is considered to be negligible. We added the following explanation to the method section:

Page 17, line 293: *"Although the epon-embedded specimens were likely to be deformed by shrinkage of the section¹⁴, we believe that it did not affect the averaging of the cryo tomograms because the section-derived maps were used only in the first iteration and subsequently replaced with newly generated averages of cryo tomograms."*

line 266: What values of CTF were obtained from the CTF determination? Did they fall within the selected range? Provide the range of determined values.

Authors' reply: We added the defocus information as below:

Page 19, line 328: *"The estimated defocus values were within the range of 1.1 – 2.0 μm ."*

II. 488-489. Given the thickness of the specimen and the tilt increment, I think that a value of 31-32 Å is overly optimistic and simply illustrates the inadequacy of the FSC as a measure of resolution. Since it is not critical to their result, I would leave it out.

Authors' reply: We added a new section to discuss the resolution issues (Pages 14-15).

Figure S1. I find it interesting that the myofibrils, which are unconstrained, only shortened a limited amount in calcium+ATP. Had they been skeletal muscle myofibrils, they would hypershortened and destroyed the structure. To what do the authors attribute the fact that the myofibrils only shortened a small amount when activated.

Authors' reply: As shown in the micrographs (Fig. 1 and S1A), we did not observe hypershortening or destruction of the cardiac myofibrils by Ca+ATP treatment. The difference in the response to Ca+ATP treatment between cardiac and skeletal muscle is supposedly related to the sarcomere length-tension relationship (Gordon et al 1966; Sonnenblick and Skelton 1974). When the sarcomere length of cardiac muscle falls outside the range of 2.0-2.2 μm , the tension drops steeply. On the other hand, the skeletal muscle exerts relatively constant tension over a wide range of sarcomere length.

Figure 5. The conformational change in alpha-actinin is displayed as a change in F-actin binding. To show the change in the alpha-actinin, it would be better to show the two structures displayed overlaid on each other without the F-actin. Since they are talking about conformational changes, a movie morphing between the two would be quite informative.

Authors' Reply: We created the supplemental movie 5 morphing the two states of the actinin dimer.

REVIEWERS' COMMENTS:

Reviewer #1 (Remarks to the Author):

The authors made a notable effort to address comments/suggestions and to clarify issues. From the amended manuscript now clear that actinin was modelled as a rigid body due to limited resolution, and the shortcomings due to this stated, e.g. "positions of the actinin and the F-actin do not necessarily reflect the actual binding".

Nevertheless, the authors inserted additional movie (5SI), which shows morphing the two conformational states of the actinin dimer, which is somewhat contradictory, as the authors state on Page 7, line 99: "The resolutions of the α -actinin densities were sufficient for rigid-body docking of the crystal structure". Additionally, the longitudinal sliding of actinin subunits as illustrated in the movie is unlikely, given the finely tuned electrostatic complementarity of the surfaces engaged in dimerization (Ylaenne et al, Structure 2001). Structural data on α -actinin suggest the rod is rigid and inflexible, but flexible segments link the actin- and titin-binding domains at the N- and C-termini.

Regarding the ABD-wt bound on F-actin: the ABD-wt have lower affinity compared to several disease mutants, where the mutations increase/facilitate the CH-CH domain opening required for binding, and which were used in the high-resolution study by Iwamoto et al 2018, cited also in the manuscript.

The authors conclude, that the wild-type ABD of actinin, which displays weak density, binds in a conformationally heterogeneous mode to F-actin:

Page 11: "Although high resolution structures of F-actin decorated with the actin binding domains has been reported 32, 33, the actin binding domains used in these studies have point mutations, which increase the affinity of the domain to the F-actin and probably stabilize the complexes. The structure of F-actin decorated with wild-type actin binding domains is low in resolution and its structural details are blurred, probably due to the structural heterogeneity 32. We suppose that the actin-binding domain of the α -actinin could bind to a wide area of the F-actin surface because the Ca⁺ATP treatment displaced the α -actinin along the surface of the F-actin (Fig. 5C, right)." This is only one possible interpretation of the weak electron density, the other can simply be that since the occupancy of ABD on F-actin is not 100% (due to low affinity or loss of decoration upon freezing in *in vitro* experiments), consequently the electron-density is weak(er). Given the fact that molecular recognition and specificity are the basis of macromolecular interactions, I am still not convinced by the interpretation of heterogeneous binding on "a wide area of F-actin surface". Furthermore, neither of the two cited articles talks (Iwamoto et al 2018, Avery et al, 2017) about the heterogeneous binding of ABD to F-actin, but about the heterogeneous conformations of ABD when not bound to F-actin, being in two states - predominantly open and closed. Additionally, the apparent dissociation constant for plectin ABD is 70 μ M (Avery et al, 2017), which is one order of magnitude weaker than that of α -actinin ABD (4.7 μ M, Way et al, 1992), while the reported affinity of filamin ABD and its mutants are 7 and 4 μ M, respectively (Iwamoto et al, 2018). It is thus difficult to reconcile the binding affinities with the proposed heterogeneity of α -actinin binding to F-actin and the known reported apparent affinities of ABDs and their mutants.

I would therefore like to invite the authors to revise the interpretation of weak ABD electron density as heterogeneous binding on "a wide area of F-actin surface" in order not to give a misleading message based on "narrow" interpretation of their and published data.

Reviewer #2 (Remarks to the Author):

On the whole we are satisfied with the response of the authors to our comments.

However we don't agree with the new statement:

Page 9, line 152: "In the previous studies based on the ultra-thin section observations, it has been

suggested that the actinin cross-bridges between F-actins of the same polarity under the rigor condition or in the small square form 7, 9.

This sentence is incorrect. Neither Goldstein nor Morris suggest in their papers that alpha-actinins link F-actins of the same polarity. The basis of the small-square lattice appearance in transverse view is that the orthogonal disposition of the ABD in relation to the rod gives the impression of same-polarity linking while the links are physically between actins of opposite polarity.

Reviewer #3 (Remarks to the Author):

Summary: This revised manuscript describes a new reconstruction obtained from cryoelectron tomography of the Z-disk from vertebrate cardiac muscle. It is a ground breaking report. As far as my comments on the original, I am satisfied with the changes. IMHO I believe the response to the other reviewer comments is also good.

Minor comments:

l. 640. Movie legends are still a bit thin. Movie 4 could use a description of the color code since it differs from that of Movie 3. The legend says related to Figure 5 but the color code is not related. Same problem with Movie 5. Should be easy to fix.

Responses to referees

REVIEWERS' COMMENTS:

Reviewer #1 (Remarks to the Author):

The authors made a notable effort to address comments/suggestions and to clarify issues. From the amended manuscript now clear that actinin was modelled as a rigid body due to limited resolution, and the shortcomings due to this stated, e.g. “positions of the actinin and the F-actin do not necessarily reflect the actual binding”.

Nevertheless, the authors inserted additional movie (5SI), which shows morphing the two conformational states of the actinin dimer, which is somewhat contradictory, as the authors state on Page 7, line 99: “The resolutions of the a-actinin densities were sufficient for rigid-body docking of the crystal structure”. Additionally, the longitudinal sliding of actinin subunits as illustrated in the movie is unlikely, given the finely tuned electrostatic complementarity of the surfaces engaged in dimerization (Ylaenne et al, Structure 2001). Structural data on a-actinin suggest the rod is rigid and inflexible, but flexible segments link the actin- and titin-binding domains at the N- and C-termini.

Authors' reply: We added a following sentences:

Page 10, line 172: “In addition, the sliding motion between the rod domains of the α -actinin is thought to be an artifact of rigid-body docking of the crystal structure. The rod domains are supposed to be rigid and inflexible.”

Regarding the ABD-wt bound on F-actin: the ABD-wt have lower affinity compared to several diseases mutants, where the mutations increase/facilitate the CH-CH domain opening required for binding, and which were used in the high-resolution a study by Iwamoto et al 2018, cited also in the manuscript.

The authors conclude, that the wild-type ABD of actinin, which displays weak density, binds in a conformationally heterogeneous mode to F-actin:

Page 11: ““Although high resolution structures of F-actin decorated with the actin binding domains has been reported 32, 33, the actin binding domains used in these studies have point mutations, which increase the affinity of the domain to the F-actin and probably stabilize the complexes. The structure of F-actin decorated with wild-type actin binding domains is low in resolution and its structural details are blurred, probably due to the structural heterogeneity 32. We suppose that the actin-binding domain of the a-actinin could bind to a wide area of the F-actin surface because the

Ca+ATP treatment displaced the α -actinin along the surface of the F-actin (Fig. 5C, right)."

This is only one possible interpretation of the weak electron density, the other can simply be that since the occupancy of ABD on F-actin is not 100% (due to low affinity or loss of decoration upon freezing in in vitro experiments), consequently the electron-density is weak(er). Given the fact that molecular recognition and specificity are the basis of macromolecular interactions, I am still not convinced by the interpretation of heterogeneous binding on "a wide area of F-actin surface".

Furthermore, neither of the two cited articles talks (Iwamoto et al 2018, Avery et al, 2017) about the heterogeneous binding of ABD to F-actin, but about the heterogeneous conformations of ABD when not bound to F-actin, being in two states - predominantly open and closed. Additionally, the apparent dissociation constant for plectin ABD is 70 μ M (Avery et al, 2017), which is one order of magnitude weaker than that of α -actinin ABD (4.7 μ M, Way et al, 1992), while the reported affinity of filamin ABD and its mutants are 7 and 4 μ M, respectively (Iwamoto et al, 2018). It is thus difficult to reconcile the binding affinities with the proposed heterogeneity of α -actinin binding to F-actin and the known reported apparent affinities of ABDs and their mutants.

I would therefore like to invite the authors to revise the interpretation of weak ABD electron density as heterogeneous binding on "a wide area of F-actin surface" in order not to give a misleading message based on "narrow" interpretation of their and published data.

Authors' reply: We removed the whole paragraph stating "wide binding surface on the F-actin" (page 11, line 176-178) because we cannot state meaningful interpretation of the actin-binding site considering the concern presented by the reviewer.

Reviewer #2 (Remarks to the Author):

On the whole we are satisfied with the response of the authors to our comments.

However we don't agree with the new statement:

Page 9, line 152: "In the previous studies based on the ultra-thin section observations, it has been suggested that the actinin cross-bridges between F-actins of the same polarity under the rigor condition or in the small square form 7, 9.

This sentence is incorrect. Neither Goldstein nor Morris suggest in their papers that α -actinins link F-actins of the same polarity. The basis of the small-square lattice appearance in transverse view is that the orthogonal disposition of the ABD in relation to the rod gives the impression of same-polarity linking while the links are physically between actins of opposite polarity.

Authors' reply: We removed the sentence and rephrased the paragraph as follows:

Page 9, line 150: "As reported in the previous thin section tomography study of the Z-disc in the small square form²⁴, the actinin cross-bridges were formed between F-actins of opposite polarity in both Ca+ATP and EGTA+ATP states (Fig. 3). A similar model has been proposed as a "diagonal square" form²⁶⁻²⁸. It is possible that our "diamond-shaped" form corresponds to, or related to the "diagonal square" form, which is thought to be different from the small square form. It would be necessary to conduct a cryo-FIB tomography^{29,30} of native muscle fibers to make a clear distinction among these forms."

Reviewer #3 (Remarks to the Author):

Summary: This revised manuscript describes a new reconstruction obtained from cryoelectron tomography of the Z-disk from vertebrate cardiac muscle. It is a ground breaking report. As far as my comments on the original, I am satisfied with the changes. IMHO I believe the response to the other reviewer comments is also good.

Minor comments:

1. 640. Movie legends are still a bit thin. Movie 4 could use a description of the color code since it differs from that of Movie 3. The legend says related to Figure 5 but the color code is not related. Same problem with Movie 5. Should be easy to fix.

Authors' reply: We added color descriptions to the movie legends.